# The Effect COVID Has Had on the Wants and Needs of Children in Terms of Play: Text Mining the Qualitative Response of the Happen Primary School Survey with 20,000 Children in Wales, UK between 2016 and 2021

**DOI:** 10.3390/ijerph191912687

**Published:** 2022-10-04

**Authors:** Michaela James, Mustafa Rasheed, Amrita Bandyopadhyay, Marianne Mannello, Emily Marchant, Sinead Brophy

**Affiliations:** 1Data Science Building, Faculty of Medicine, Health and Life Science, Medical School, Swansea University, Swansea SA2 8PP, UK; 2Play Wales, Park House, Greyfriers Road, Cardiff CF10 3AF, UK

**Keywords:** COVID, play, health, wellbeing, children

## Abstract

Play is central to children’s physical and social development. This study examines changes in children’s response to questions on play opportunities between 2016 and 2021. Primary school children aged 8–11 in Wales participated in the HAPPEN survey between 2016 and 2021. The survey captures a range of information about children’s health and wellbeing, including open-ended questions about what could make them happier. Text mining methods were used to examine how open-ended responses have changed over time in relation to play, before and, after the COVID enforced school closures. A total of 20,488 participant responses were analysed, 14,200 pre-school closures (2016 to pre-March 2020) and 6248 after initial school closures (September 2020–December 2021). Five themes were identified based on children’s open-ended responses; (a) space to play (35%), (b) their recommendations on play (31%), (c) having permission to play (20%), (d) their feelings on health and wellbeing and play (10%) and (e) having time to play (4%). Despite differences due to mitigation measures, the predominant recommendation from children after COVID is that they would like more space to play (outside homes, including gardens), more time with friends and protected time to play with friends in school and at home.

## 1. Introduction

Play is central to children’s physical, mental, social and emotional health and wellbeing and helps develop skills such as problem-solving, communication, fine motor skills and confidence [1,2,3]. There is also importance placed on the role of play in developing physical literacy. This is defined as the motivation, confidence, competence, knowledge and understanding to value and engage in physical activities [4]. Thus play equips children with the skills to sustain an active lifestyle into adolescence and adulthood [5]. The importance of play has been recognized by the United Nations Convention on the Rights of the Child (UNCRC), where play has been enshrined under Article 31 which calls for children to be able to participate fully and equally in recreation and leisure activity. It also calls for them to have a right to be heard and taken seriously on all matters affecting them (Article 12) and to gather and use public space (Article 15) [6]. This paper takes its definition of play from the United Nations Committee on the Rights of the Child’s General Comment 17 on Article 31 of the United Nations Convention on the Rights of the Child [6]. It defines play as a behaviour, activity or process initiated, controlled, and structured by children, as non-compulsory, driven by intrinsic motivation, not a means to an end and that has key characteristics of fun, uncertainty, challenge, flexibility, and non-productivity.

Play during childhood has positive impacts on multiple important long-term health outcomes including increased physical activity, improving wellbeing in children, and helping to develop resilience [7]. It is also crucial and worthwhile for the enjoyment it brings to children and their families in the moment [8,9]. Play is fundamental for good health and wellbeing; for example, being physically active through play supports children physical and emotional development, contributing to their health and wellbeing [10]. When they play, children contribute to their immediate wellbeing and to their own development.

In March 2020, the World Health Organisation declared a global coronavirus pandemic (COVID-19) caused by the transmission of severe acute respiratory syndrome coronavirus 2 (SARS-CoV-2) [11]. Countries worldwide, including in the United Kingdom implemented a range of public health measures and mitigation strategies to reduce transmission, including a strict ‘stay at home’ policy and a national lockdown running (March to June 2020 and December 2020 to January 2021). This meant the closure of many educational, workplace and retail settings with social gatherings all but banned outside of direct household members [12]. The implications of the restrictions had the potential to directly impact play opportunities for children due to the lack of outdoor access, closure of play spaces, time with friends, school closures and online learning [13,14,15]. However, while pandemic research has highlighted decreases in activity in children [16,17,18], there is also evidence to suggest that the pandemic may have been a good opportunity for children to play and be active [19,20].

Using open-ended responses from the HAPPEN-Wales survey regarding what children wanted to make them happier and healthier, this study aims to: examine how children’s reports of play have changed pre and post-COVID and to provide key recommendations on COVID recovery plans for children based on the perspectives of children.

## 2. Materials and Methods

HAPPEN-Wales is a primary school network [21] that was established in 2015 following research with primary school headteachers who advocated for a more cohesive approach to prioritising health and wellbeing in schools. They advocated for bringing together schools and partners in health and research to provide evidence to make more targeted plans for schools based on the needs of their pupils [22,23]. To address this, HAPPEN co-produced the HAPPEN Survey, an online self-report questionnaire that was developed and designed alongside children, teachers and stakeholders in education [21]. The survey is completed by primary school-aged children (aged 8–11) in the school setting and captures a range of self-reported health behaviours including physical activity and sedentary behaviour, diet and dental health, wellbeing and mental health and the local community. Once complete, schools receive an individual report of pupils’ group-level health and wellbeing data compared to national averages. This school report is aligned to the new Curriculum for Wales (due to be rolled out in 2022) [24], whereby Health and Wellbeing is one of six distinct curriculum areas, enabling schools to tailor their health and wellbeing plans and curriculum development based on pupils’ needs.

### 2.1. Participants

Since 2016, primary schools have been invited to take part in the HAPPEN Survey throughout the academic year. Participation for both schools and children is voluntary, and they have the right to withdraw at any time. Schools are invited to share details of the survey (including study aims and a parent information sheet) among parents/guardians so that parents were given the opportunity to opt their child out from the survey. This opt-out method of recruiting participants was introduced in 2019 and aimed to ensure that a representative sample was recruited to reflect all children in Wales. Child assent is also obtained at the start of the survey.

### 2.2. Data Collection

As part of the HAPPEN survey, children are asked “*What could be done to make you happier and healthier in your local area?”.* This is an open-ended question with the aim of understanding what children want and need from their local communities. This question has been asked since the survey’s development in 2016 and has over 20,000 responses. We analysed the responses that related to play to examine how play has changed through COVID from pre-pandemic mitigation measures (2017–March 2020) to during/post-COVID mitigation measures (September 2020–December 2021). These time periods have been defined as pre and post by the nature of school closures at the time (Viner et al.). In March 2020, schools were required to close in Wales as part of transmission mitigation measures. Schools reopened in June as part of phased return before opening in September 2020. A period of school closures was then enforced in November 2020 until January 2021. For this study, pre-pandemic is the period prior to March 2020 with during/post-pandemic running from June 2020 to December 2021.

This open-ended question was only asked when children were attending school face-to-face, therefore during school closure periods we did not collect this information. The most recent version of the HAPPEN Survey can be seen as Appendix A. The process of data coding involved two researchers. The first researcher downloaded the raw data, cleaned the data, checked for duplicates, generated a unique participant ID number, and removed identifiable information (MJ). This process protects participants’ anonymity by ensuring that the second researcher coding the responses and conducting the analysis could not identify individuals (MR).

### 2.3. Analysis

Initially, a random sample of 1000 responses were selected for qualitative thematic analysis to identify common themes from the open-ended responses to the HAPPEN survey. This was led by MR who identified common key words, frequently repeated words, and iterations of those words. These frequent words were sorted into a lookup dataset used as a reference for text mining. Opportunities for children to play can be supported or restricted through having time (children’s ‘free’ time when they can become immersed in playing), space (how public space can support or constrain children’s ability to play as well as access to designated spaces for play) and permission to play (children’s subjective experiences of time and space, including factors such as a sense of freedom, permission, belonging, fear and harassment, as well as the increasing adult appropriation and control of play) as stressed in Wales—A Play Friendly Country statutory guidance to Local Authorities [2]. Therefore, this was used to underpin coding. For example, mentions of parks, gardens, and playgrounds (e.g., “*we could get a park nearer to our house*” or “*a bigger garden*” were coded within spaces to play). Responses were removed from analysis if they were left blank by the participant or did not discuss play (*n* = 4217).

These initial codes were used to identify themes using text mining methods (see details below) on the whole dataset of responses. The text mining involved three stages [25,26]. Stage 1, or the pre-processing stage, included tokenization (breaking the text into tokens or small sentences such as words), removing stop words and, lemmatization (identifying the base form of the word, e.g., good is the base form of better). Stage 2 involves a frequency analysis of words (e.g., in our dataset the word park is mentioned >1000 times, friends is mentioned >1000, litter is mentioned >400 and the word combination feeling safe is mentioned >300 times) followed by manual review, with an expert review of the words to ensure none are missed and only relevant words are considered. Stage 3 involves co-occurrence analysis to identify words that are associated together (e.g., play & park), identify word pairs that should be coded together, synonyms (e.g., park and parc should be coded park), link to the data and code all the pair words.

The quality of coding by the automated text mining method was compared by two researchers (MJ & MR) and the text mining method was modified to improve accuracy. Following this second round of coding, key themes were identified which include (i) Time to play, (ii) Space to play (access to play, safety when playing, having space to play, sustainability) (iii) Permission to play (being allowed to play, having relationships that permit play), (iv) Recommendations (specific play/activity recommendations) and, (v) Health and Wellbeing and play (how play makes children happier, how play can be helped with healthy diets). These themes were then stratified by the period of pre-COVID and post-COVID.

## 3. Results

A total of 20,488 participants completed the HAPPEN Survey between September 2016 and December 2021 (*n* = 46% boys, average age = 9.97). From these responses, a total of 16,271 responses were coded which discussed play. Of this number, 12,529 responses were from 2016–March 2020 (pre-pandemic, *n* = 23%) with the rest from September 2020–December 2021. Across both time points, the codes showed that children discussed; space to play (35%), their recommendations on play (31%), having permission to play (20%), their feelings on health and wellbeing and play (10%) and having time to play (4%). Figure 1 shows the most frequent coded words in a word cloud.

Table 1 shows a breakdown of this stratified by pre and during/post-COVID mitigation measures, with no significant difference in the priorities of children between time points.

From these initial codes, additional sub-themes were identified; (i) access to spaces to play, (ii) perceived safety of local spaces, (iii) the need to improve existing spaces, (iv) cleaner spaces, (v) being allowed to play, (vi) relationships that facilitate play, (vii) health and wellbeing implications of play and, (viii) having more time to play. This can be seen in Table 2 and again, highlights (aside from the emergence of specific COVID responses) there were no significant differences in their occurrence pre and during-COVID.

A breakdown of key themes and quotes can be seen as Appendix A.

### 3.1. Specific Recommendations for Play

Throughout the responses, there were many recommendations made by children for play and sport-based activities which were grouped under the specific recommendations theme to help provide clear examples of the diverse range of activities suggested. This reflects the range in children’s wants and needs from play. Some common suggestions for activities were swimming, basketball, and football. Although lots of children asked for the opportunity to do “different” activities:

“Go swimming more” (2016/2017)

“Swings in the park, better football pitch, somewhere to play basketball”(2019/2020)

“To run with my friends more and play different activities.”(2020/2021)

The school setting was cited as important for this with some children saying they wanted *“more PE”* or *“more after school sports”.* This theme suggests that time allocated for school-based activity needs to be dedicated to a broad range of activities. The recommendation from this theme is to consult children (e.g., through pupil voice groups at school, community groups outside school) to identify specific wants and needs from activities and play as this would be different in different settings. For content, Figure 1 highlights how prominent the school setting is within responses and how frequently mentioned sport, exercise, equipment, and more specific forms of activities are mentioned.

### 3.2. Space to Play

Space to play was a significant code across all academic years and pre/post-pandemic. Much of the discussion revolved around the theme of improving existing spaces (18% and 19% respectively), and improving play equipment, fixing damage and cleaning local parks:

“Make sure that parks and other places have safe equipment”(2018/2019)

“Fix damage to parks by me”(2021/2022)

“Have a litter pick up once a week…”(2017/2018)

This suggests that what currently is on offer does not meet the standard of children’s wants and needs and to enable them to play more, parks need to be improved. This shows that despite infrastructure in place, more needs to be done to ensure that it is actually usable. Figure 1 Shows that parks, halls, environments (generally speaking), gyms and the indoors are mentioned often as spaces to play.

Within this code, safety of spaces was highlighted. Children consistently mentioned safety concerns and fears in some of the places they would like to play including fears of illegal behaviours (e.g., drug use) as well as recommending safety equipment checks. This represents children’s awareness and fears regarding illegal behaviours in their local area, either through direct observation or indirectly by adults or observed directly by them.

“There are unfriendly people hanging around my area doing drugs and smoking”(2018/2019)

“More safer and clean areas for children to play and feel more comfortable!”(2021/2022)

Some children also noted the presence of bullying and “unkind” behaviour which deters them from playing. The word ‘nasty’ is mentioned in Figure 1. Concerns over safety also highlighted the presence of cars:

“My road is very busy so we could get some more traffic lights.”(2016/2017)

“Put speed limits on the roads”(2016/2017)

Children even made clear suggestions about how to improve the safety of their local areas with the mention of traffic flow measures. It is evident they are aware that local communities prioritise the use of cars over the safety of pedestrians. They also note the presence of litter, including *“dog poo”* which appears to be a clear deterrent to play.

“Litter and dog poo and speeding”(2016/2017)

“Put more bins so that there is less litter.”(2017/2018)

Access was mentioned in terms of linking homes with facilities better including better infrastructure for children to be able to get to spaces.

“We could get a park nearer to our house.”(2016/2017)

“Easier to access outdoor places”(2021/2022)

It is interesting that a significant proportion of responses discuss limited accessibility to outdoor spaces. Responses to the general spaces code highlight possible reasons including poor equipment, fears over safety and cleanliness. Interestingly, in 2021 more responses mentioned their garden as a space to play (e.g., “A bigger Garden.”) which would have been in line with the emergence of COVID and subsequent restrictions seeing children needing to access play in their household. It is worth considering that some children will not have had a garden and therefore, may have had no space to play at all.

The key recommendation from this theme is to not only provide outdoor spaces that children can easily access play in local communities but also to provide spaces to listen to the concerns of children and facilitate ways in which these fears can be reduced.

### 3.3. Permission to Play

Permission to play was centered around relationships that permit play and how relationships with friends and family are conducive to giving implicit/explicit permission for children to play. In terms of family, the presence of parents was key. Children note how parents living separately and the business of parents is not conducive to their play. This highlights how parental figures can be role models and leaders for play in certain spaces, their time gives permission for children to play:

“Do more sport with my mum”(2016/2017)

“Mum and dad to live with each other”(2018/2019)

“To have my mum not be so busy” (2021/2022)(2021/2022)

Having friends to play with was considered essential throughout the years including having more live nearer, “being around” friends more and having opportunities to go out with them. Under friendship, bullying was consistently mentioned as something that deterred play with the word “kind” mentioned frequently in reference to friendships and bullying behaviour:

“Stop bullying”(2019/2020)

“Be kind to people and make others to be kind to others”(2021/2022)

This also has parallels to safety concerns in the space code. Under this theme, it is evident that social, supportive and “kind” environments give permission to play. Therefore, the key recommendation from this theme is to facilitate opportunities for children to be with their friends and family where possible.

### 3.4. Health and Wellbeing Outcomes

The theme of health and wellbeing outcomes relating to play emerged in the academic year 2018/2019 which is in line with the announcement of the new Curriculum for Wales in which health and wellbeing is one of six distinct areas. This suggests that children had a greater awareness of health and wellbeing after this year, particularly how this would impact and be impacted by their play:

“eat more veg and stay fit and healthy (ALWAYS)”(2019/2020)

“Keep fit always get fresh air”(2020/2021)

This shows that children not only want to play, but acknowledge the benefits of being able to play and the benefits that behaviours such as healthy eating could have for play. It is also interesting that some children expressed a need to socialise and awareness of the importance of this to them. Some of the most frequently mentioned words in terms of health and wellbeing can be seen in Figure 1, particularly fruit, vegetables and healthy.

### 3.5. Time to Play

The theme of time to play centered around children wanting more time to play and, in terms of more time to play, this was time outside playing or more opportunities to play/be active. As with all themes, there was no significant difference between pre and post pandemic responses, just that more time was needed.

“Spending more time swimming and going outside”(2016/2017)

“Allowed to play out all the time because we like playing out”(2019/2020)

“To have more time to play”(2021/2022)

While there were implications that this meant more time to play at home, there were explicit mentions of how the school setting mitigated time to play. In particular, there were many mentions of more break times and afternoon play as well as the mention of homework being done in free time. This theme is also the only theme that mentions how time to play impacted genders differently, with girls saying there was a lack of time dedicated to activities that they liked:

“Not just boy sports at play times so girls can play as well.”(2018/2019)

While device use was mentioned throughout the academic years, it becomes particularly prominent in post-pandemic. This could be because of online learning during school closures and the use of electronic equipment required. The key recommendation from this theme is to protect play time and where possible facilitate it as much as possible. This is very relevant in the school setting where break times are often used as a behaviour management technique or where afternoon breaks have been removed in favour for increased learning time.

### 3.6. The Impact of COVID

In 2020, children started discussing COVID-19 and the impact this was having on their ability to play and be happy. For the most part, this revolved around “stopping” transmission where children acknowledged that this needed to be done. However, when taken in conjunction with other responses:

“Get rid of COVID rules”(2021/2022)

“seeing each other a lot more but because of COVID we have not been doing that”(2021/2022)

“Easier interaction not staying away from each other (COVID)”(2021/2022)

It is evident that stopping the transmission equated to the end of the restrictions for children. These restrictions impacted socialisation and children specified that they wanted to see their friends again and not have to socially distance. It is interesting to note that children did not discuss COVID-19 in relation to the impact on their health showing that this age group was not concerned about contracting the virus. They were more concerned about the impact this was having on the ability to see their friends. The recommendation from this theme is to acknowledge that children have missed key socialisation time and to facilitate this and nurture this in recovery plans. Play is an important place for this to happen.

## 4. Discussion

This study aimed to examine how children’s reports of play have changed pre- and post-COVID and to provide key recommendations that can inform COVID recovery plans based on the perspectives of children. There is a well-established body of solid evidence that shows the contribution that play, particularly self-organised play, can make to children’s immediate and long-term wellbeing, to their physical health, problem-solving, communication, fine motor skills, confidence and to their mental health and resilience [7,27]. Therefore, it is vital we protect play and listen to the best ways in which to facilitate it.

Ten key themes were identified under the codes of Time (time to play), Space (access to play, safety when playing, having space to play, sustainability), Permission (permission to play, relationships and play), Recommendations (specific play/activity recommendations), Health and Wellbeing Outcomes (health and wellbeing and play) as a result of analysis of children’s responses from 2016 to 2021. The text mining analysis showed that the presence of these themes was similar pre- and post-pandemic, showing that the recommendations for children’s play remain similar throughout time and circumstance.

The specific suggestions made by children were diverse and broad, encompassing a range of answers from football, basketball, and swimming to simply asking for a variety of more activities. This highlights that there is not a one size fits all model to promote activity and play in children and therefore, it is a worthwhile pursuit to engage and involve children themselves [28]. Previous research has highlighted that older children have requested more choice in activities, rather than prescriptive, adult-led programs [28]. This study shows that a similar request is being made by younger children with a wide-range of play-based and sport-based suggestions being bought forward. Therefore, we cannot assume a ‘one size fits all’ model should work with play. Other studies have acknowledged that choice, or lack of, is a reason why young people become disengaged with activity, particularly girls [29]. It is important that suggestions are acknowledged and the diverse and broad range of interests and needs are considered. This can be done so via a period of consultation with children. By asking them, interventions and programs can implement exactly what they want and need, helping to improve the success, longevity, and sustainability of play-work in communities and opportunities to play in schools.

Speaking to children about their solutions could have a significant impact on designing and implementing pandemic recovery plans and is reflected across the themes emerging in this study. While it is not always feasible to provide a wide-range of opportunities due to lack of resources (e.g., space, time or funding) [29], research highlights that providing autonomy and creating partnerships between users (in this instance, children) and those delivering (e.g., teachers, play workers) could improve the sustainability and efficiency of play/activity opportunities to improve health and wellbeing [30,31]. Particularly, in the wake of the pandemic where play was reduced due to mitigation measures, it is important we look to facilitate the best opportunities for children to overcome this lost time. To do this, we need to look at their suggestions and what previous research has acknowledged as positive play/activity-enablers.

Throughout the themes children recommend protecting spaces to play with their friends as a priority. This remained a constant throughout the time periods. Space to play was the most frequently discussed amongst children, encompassing access, safety when playing and sustainability of play spaces. This was centered around local park spaces, their upkeep and maintenance, safety, and the general accessibility to these spaces. Parks have been acknowledged by previous research to be important spaces for play [32,33,34]. Playing in these environments supports children to feel part of their neighbourhoods and wider communities, allowing children to learn about the world around them, make connections, and develop a sense of identity and belonging [1]. Thus, this access is integral for their development and lived experience (or enjoyment). Access could mean closer proximity [34] but it could also refer to access to features within the park such as open spaces, courts or trees to climb [32]. Fears over safety may reduce access [35,36] due to concerns over encountering illegal behaviours and unsafe equipment/environments as mentioned by children in this study. Independent mobility in children and the ability to explore, play and be active in local outdoor spaces without adult supervision has declined [37,38]. This needs to be protected and work needs to be done to improve mobility for children in their local neighbourhoods.

It is vital that we listen to the concerns of children and facilitate ways in which these fears can be reduced, and access improved. Previous research has shown that perceptions of safety are significant in facilitating play/physical activity, particularly for those more deprived [20,35]. Concerns over safety were more frequently mentioned post-2019 where, in 2020, children would have been subject to strict lockdown restrictions [12]. Studies suggest that being confined to local areas during periods of restriction may improve perceptions of safety [20].

In light of COVID-19, it is vitally important that we give children space to play due to the lack of access to outdoor spaces, time with friends and school closures [13,14,15] that they have faced as a result of mitigation measures. During the pandemic and its associated lockdowns, access to outdoor play was particularly important as the population was confined to their homes. Outdoor play gives sense of freedom and control that children can enjoy and it allows for children to be energetic and physically active. Play is a form of physical exercise for children. However, to stop transmission, playgrounds were closed during lockdown. In Wales, as more was learned more about the pandemic and ways to manage transmission, Welsh Government advice was updated to establish the important role that playing has in supporting wellbeing. The government provided guidance and prioritised the opening of parks, playgrounds and childcare and play work provision from summer 2019 onwards, highlighting the need to play. This need remains, and as children have stated, the protection and maintenance of play spaces should still be high on the agenda. A recent study of children’s experiences of playgrounds in the pandemic highlights this further with children expressing their pleasure at being back in the playground with no social distancing [39].

Alongside space, children discuss the role of relationships in supporting play opportunities. Play supports socialisation. As mentioned above, children have welcomed the removal of social distancing in facilitating socialisation [39]. Our study highlights children value the characteristics of kindness in their peers and concerns about bullying were mentioned suggesting children seek nurturing and supportive structures which will facilitate their play. In 2019, Welsh Government launched new guidance on how to stop bullying in schools [40]. Its consistent mention suggest work still needs to be to eradicate bullying from the school setting. Discussing and calling out bullying can support children to develop confidence to play interdependently outside of school.

With family members, it was apparent that children see parents/caregivers as role models and supporters for play. This is important to note as previous research has highlighted the novelty of the pandemic enabling some children to spend extra time with family members that they otherwise would not have had due to an increased number of parents working from home or being furloughed [20]. This has also been acknowledged by school staff who reported children having more opportunities for walking and, spending time outside, with this contributing to strengthened family relationships [18]. This may then have had a positive impact on some children during this time and this time should be valued.

The pandemic saw children access their local community for exercise and play during the pandemic, mostly being accompanied by parents [41]. Care must be taken to ensure that, as restrictions on outdoor spaces is relaxed, that children feel connected to their communities to help them to gain confidence to play out. For children, their main concern with the pandemic were restrictions on socialisation and having to distance from their friends. Research has shown that these measures will have had an impact on children’s wellbeing [42] with the removal of positive interactions with peers, teachers, coaches and wider family members. Therefore, as part of COVID-19 recovery plans, it is essential we value that children need this socialisation time back. With the uncertainty caused by the pandemic, opportunities to play are vital to helping children make sense of their experiences, problem-solve, reconnect with their peers, and promote their own wellbeing. As we develop interventions and initiatives to support children emerging from the pandemic and its related restrictions, play is one of the most important areas of focus to promote children’s health and wellbeing.

The wider perception and impact of lockdown for children is mixed in the literature. There is research to suggest that lockdown may have been a positive experience for young people, with physical activity improving, as well as sleep and overall wellbeing for some children (those less deprived) [20]. Adult perceptions suggest physical activity decreased, with wellbeing also at risk during this time [18]. Therefore, there is some contention on what lockdown has meant for children. Yet, from a children’s perspective, play recommendations remain the same with them asking for more variety of activities, more space, more time and more opportunities. COVID specific responses from children show that they were aware of mitigation measures and the impact these were having on their play and socialisation. Despite being deemed the population least vulnerable to direct harms from COVID infection, the government enforced restrictions impacted the lives of children in an unprecedented way [20]. Many restrictions were implemented without consultation with children or young people, and it is still unclear what the longer-term impacts of measures such as school closures have been on children. Children were specifically concerned about the impact this was having on the ability to see their friends. Protecting play, socialisation, and opportunities to be active is paramount in recovery plans and has been reflected in other studies in this field [18,20].

The above recommendations have come as a result of analysing children’s responses and are therefore advocated and suggested by children themselves. It is evident that protecting spaces to play (including investment in maintenance, upkeep, and safety) and, facilitating opportunities for children to be with their friends are important to children to help them play. These have remained constant themes throughout time and therefore, it is evident that while these recommendations are not new learning, more needs to be done to influence policy, decision-making and funding into putting these into practice.

### Limitations

This study encompasses responses from 2016 to December 2021. HAPPEN rolled-out to wider Wales in 2018 however, we cannot ensure a fully representative sample of children has been recruited across Wales. While the sampling strategy was the same for 2018–2020, data were sampled more purposefully in earlier years from South Wales which may have an influence on findings from this year. Although all schools in Wales were contacted with details regarding the HAPPEN Survey, the findings in this study only represent the views of children who took part. The perspectives captured in this study may not account for the full breadth of lived experiences of all children.

## 5. Conclusions

Drawing upon children’s responses, it is essential that we advocate for the wants and needs of children particularly in relevance to giving children broad opportunities to play and be active, space to play (particularly in reference to safe and accessible spaces designed with children in mind), facilitating socialisation (especially in light of social distancing measures), and acknowledging how beneficial and integral play is the development, health, and wellbeing of children. We cannot overlook the importance of play. It has and always will be important to protect play. Given the importance of play for children at times of crisis, recognising this during and while emerging from a global pandemic could be an enormous step forward in terms of protecting the mental health and wellbeing of our children, and of future generations.

## Figures and Tables

**Figure 1 ijerph-19-12687-f001:**
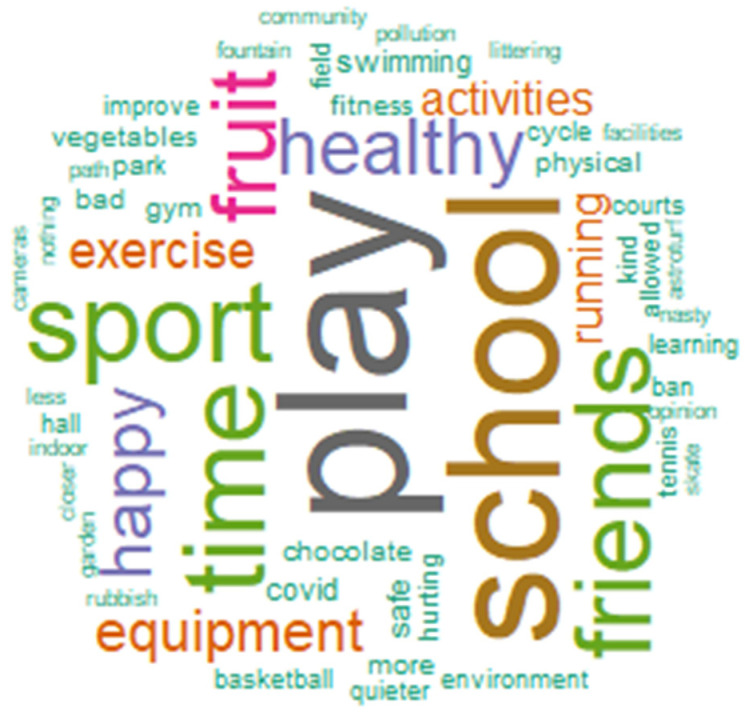
Frequently coded words.

**Table 1 ijerph-19-12687-t001:** Codes.

Codes	Pre-COVID	During/Post-COVID
Space	35%	35%
Permission	20%	20%
Time	4%	5%
Recommendations	32%	29%
Health and Wellbeing	9%	11%

**Table 2 ijerph-19-12687-t002:** Themes.

Themes	Pre-COVID	During/Post-COVID
Access to Spaces	5%	5%
Safety	7%	8%
Improve Existing Spaces	18%	19%
Cleaner Spaces	5%	3%
Being Allowed to Play	4%	3%
Relationships that Permit	16%	17%
HWB	10%	8%
More time	3%	5%
Specific activities	32%	29%
COVID	0%	3%

## Data Availability

The data are available to the research team according to ethical approval. The corresponding author is happy to provide data if required for scrutiny.

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
