# Peer review of "The Effect COVID Has Had on the Wants and Needs of Children in Terms of Play: Text Mining the Qualitative Response of the Happen Primary School Survey with 20,000 Children in Wales, UK between 2016 and 2021"

_ijerph, 2022, doi:10.3390/ijerph191912687_

Round 1
Reviewer 1 Report
The authors aimed at roviding evidence regarding the effect covid ahs had on childrens wants and needs in terms of play. They have access to an incredible sample and provided information using text mining from an open end question. It is interesting, that there was no real change observed. However, overall, i have the feeling hat the manuscripts provides decriptives mainly, which is okay.
Maybe I missed it, but i did not see how many from the 14k answers were pre or post?
In the introduciton, it may be easier to read if play was defined earlier but in the last sentece of the introduction.
line 192. a space too much
word cloud does not have enough DPI
The only thing, where i am not agreeing is in the very end in the discussion. The authors claim that their results lead to some "key recommendation". While I agree to the recommendations, the results do not indicate that we should listen to and consult children regarding the wants and needs of children. This is more common idea of how we want to live than derived from the results of this paper. Therefore, i would suggest to reframe those
Reviewer 2 Report
he article has an interest adapted to the adaptation of educational processes in periods such as the COVID pandemic: The survey research process is adequately conducted within scientific standards.
However, the exploitation of the results is very restricted to descriptive elements on the type of play, spaces and times for it. And the findings are logical after a period of pessudo-confinement or low social interaction.
However, I believe that in the conclusions the authors should go a little further than description and adopt a comparative discussion with other works and an evaluative one, even if this entails a certain scientific risk, since this is precisely the greatest contribution that this article could make.
The needs of play do not only respond to a space and time for it, but also to psychological issues and social relations. Moreover, the pandemic has restructured these aspects. And so, for example, children now play more online or digitally, even though they already have physical spaces available to them.
Reviewer 3 Report
The authors draw attention to the impact of Covid-19 on wants and needs of children in 2 terms of play. The theoretical analysis, the research method and the description of results do not raise significant objections. The aim is formulated in the introduction: to examine how children's 61 reports of play changed pre- and pst_covid and to provide key recommendations on Covid recovery plans for children based on the perspectives of children. The authors should put the necessary emphasis on the discussion section because the 3 recommendations offered at the end of it are too general, known to everyone, and do not provide new knowledge.
Round 2
Reviewer 3 Report
Thanks